# Minimal non-abelian nodal braiding in ideal metamaterials

Huahui Qiu[1,3], Qicheng Zhang [1,3], Tingzhi Liu[1], Xiying Fan[1], Fan Zhang [2] & Chunyin Qiu [1] ✉

Exploring new topological phases and phenomena has become a vital topic in condensed matter physics and materials sciences. Recent studies reveal that a braided colliding nodal pair can be stabilized in a multi-gap system with $PT$ or $C_{2z}T$ symmetry. This exemplifies non-abelian topological charges beyond the scope of conventional single-gap abelian band topology. Here, we construct ideal acoustic metamaterials to realize non-abelian braiding with the fewest band nodes. By emulating the time with a sequence of acoustic samples, we experimentally observe an elegant but nontrivial nodal braiding process, including nodes creation, braiding, collision, and repulsion (i.e., impossible to annihilate), and measure the mirror eigenvalues to elucidate the braiding consequence. The latter, at the level of wavefunctions, is of prime importance since essentially braiding physics aims to entangle multi-band wavefunctions. Furthermore, we experimentally unveil the highly intricate correlation between the multi-gap edge responses and the bulk non-abelian charges. Our findings pave the way for developing non-abelian topological physics that is still in its infancy.

Over the last decade, the study of topological phases of matter has made fruitful achievements with far-reaching impacts. A particular example is the discovery and characterization of hundreds and thousands of topological materials protected by time-reversal and/or crystal symmetries[1-8]. Symmetry plays an essential role in distinguishing topologically inequivalent band structures[3,9]; this has recently culminated in systematic classification schemes that address the global band topology in terms of irreducible representation combinatorics and symmetry-based indicators[10-15]. The paradigm leading to this success of topological band theory is to divide energy band structure into two subspaces, i.e., an occupied subspace spanned by bands below an energy gap and an unoccupied subspace above the gap. This yields additive group structures for topological invariants when adding extra occupied bands. Physical systems captured by this framework of single-gap topology are characterized by abelian groups with commutativity and additivity, with a prominent example being the tenfold way classification rooted in the K-theory[16-18].

Very recently, symmetry-protected multi-gap band topology have been predicted theoretically[19-22]. With multiple bandgaps entangled together, the underlying new paradigm is characterized by non-abelian groups. The multi-gap topology exhibits appealing noncommutative braiding structures and reflects an entirely new category of band topology. The braiding in energy-momentum space can be described by non-abelian charges[20-22]. For example, a $PT$ or $C_{2z}T$ symmetric three-band system can host a real-valued Hamiltonian, the three real eigenstates in Hilbert space constitute an orthonormal frame[20-23], and the eigenstate trajectories along a closed path define a non-abelian frame charge for the system. According to the homotopy group theory, the charge is associated to the flag variety $SO(3)/D_2$ and described by anticommutative quaternion group[20]. Particularly, the multi-gap nodes can braid reciprocally in two-dimensional (2D) momentum space as they evolve with external parameters. The non-abelian quaternion charge of a node can change sign through braiding with another node in an adjacent gap, which can result in a pair of nodes of

[1]Key Laboratory of Artificial Micro- and Nano-Structures of Ministry of Education and School of Physics and Technology, Wuhan University, Wuhan 430072, China. [2]Department of Physics, University of Texas at Dallas, Richardson, TX 75080, USA. [3]These authors contributed equally: Huahui Qiu, Qicheng Zhang. ✉e-mail: cyqiu@whu.edu.cn

an identical charge in the same gap. The identically charged nodes will not annihilate after collision, as the key signature of a nontrivial nodal braiding process[20–22]. Besides the quaternion charges, the nodal braiding can also be characterized by Euler classes[21,24] and Dirac strings[19,25]. Non-abelian topological systems have been ingeniously realized by photonic[26–28], transmission line network[27,29] and acoustic[30–32] means. And relevant phase transitions have also been clearly observed[30]. However, unambiguous experimental evidence for the long-desired nodal braiding remains elusive, because of the extreme challenges in resolving the evolution of nodal points in energy-momentum space, demanding the extraction of wavefunction information in Hilbert space, and hence concluding the non-abelian phase transition at the ultimate level of wavefunctions.

In this work, we report an elegant acoustic realization of non-abelian nodal braiding using a simple square-lattice model, in which the braiding protocol is implemented by using a sequence of acoustic samples. Unambiguously, we observe a nontrivial nodal braiding process in energy-momentum space, benefiting from the extremely clean bulk dispersions with minimal nodes residing only at high symmetry lines (HSLs) of the Brillouin zone (BZ). Particularly, we present the first experimental evidence, at the level of wavefunctions, for the band inversion and the resultant stability transition of a colliding nodal pair at a high symmetry point (HSP), as a hallmark manifesting the

nontrivial multi-gap braiding effect. Moreover, we explore the intricate bulk-edge physics of our 2D system from the perspective of braiding 1D subsystems, and unveil the nodal braiding through a global evolution of multi-gap edge states.

## Results

### Minimal model for non-abelian nodal braiding

Figure 1a illustrates two pairs of band nodes in a 2D three-band system with $C_{2z}T$ symmetry. Each node carries a non-abelian charge[20–22] ($Q$) in the anticommutative quaternion group $G_Q = \{\pm 1, \pm i, \pm j, \pm k\}$, in which the elements satisfy the fundamental multiplication rules $i^2 = j^2 = k^2 = -1$ and $ij = k, jk = i, ki = j$. Upon introducing time-evolved system parameters, the nodes can move in momentum space and even collide somewhere (usually at a HSP). The total charge of two colliding nodes in the same bandgap is determined by their product. If the result is $+1(-1)$, the band nodes will (will not) annihilate pairwise after collision[20–22]. Here we consider a nontrivial braiding process that transfers an unstable nodal pair ($Q = +1$) to be a stable one ($Q = -1$), as sketched in Fig. 1b by the time-evolved trajectories of the gap-I nodes (blue lines) and gap-II nodes (red lines). In general, a pair of newly created band nodes (green stars) carry opposite quaternion-valued frame charges (QFCs) and their total charge is $+1$. The charge sign of a given node can be flipped via braiding with another node of an

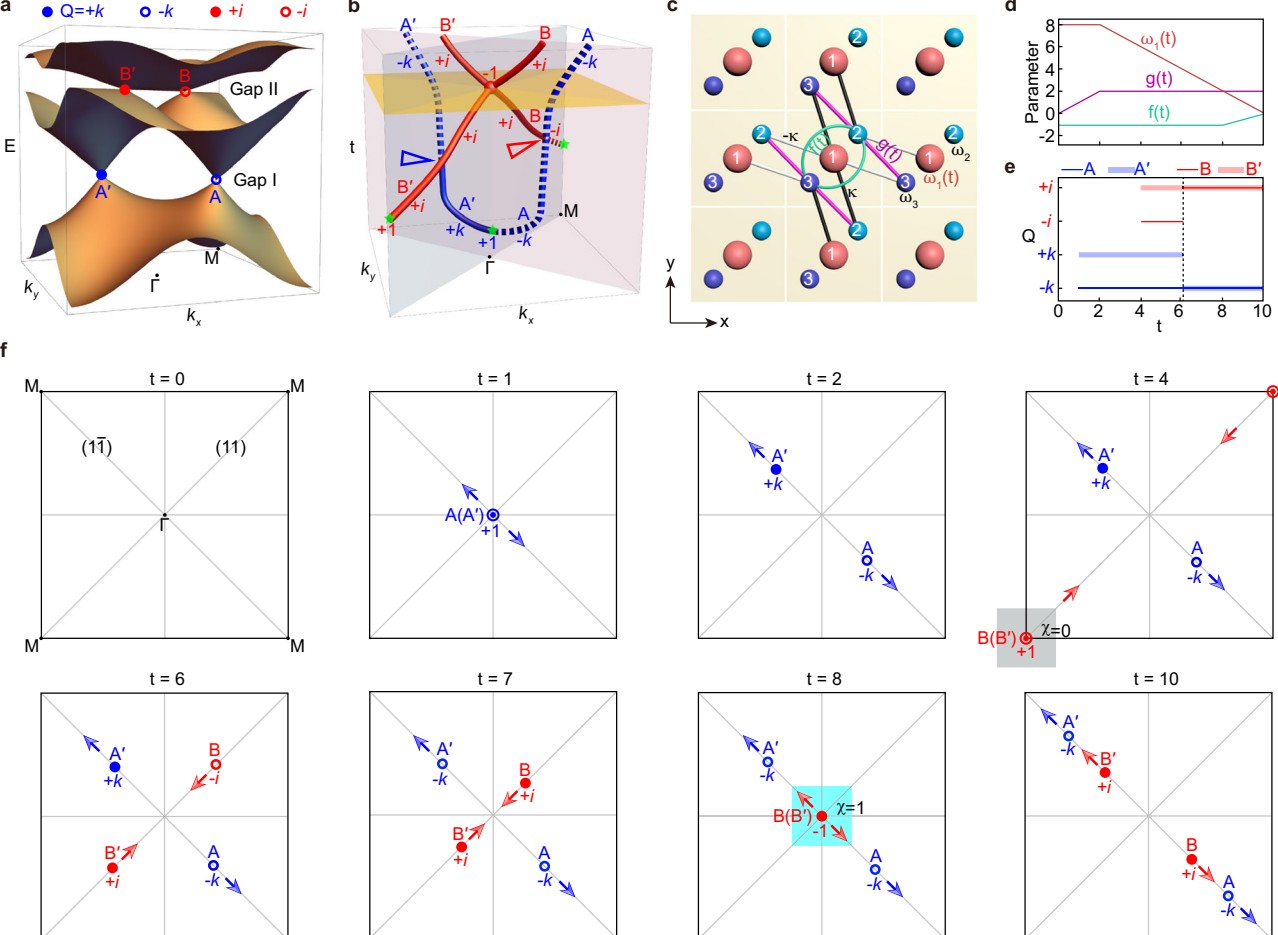

**Fig. 1 | Ideal non-abelian braiding with the fewest nodes. a** Schematic of a three-band system with two pairs of band nodes in 2D momentum space. The gap-I nodes A(A′) and gap-II nodes B(B′) carry non-abelian charges $Q = \pm k$ and $Q = \pm i$, respectively. **b** Time evolution of the band nodes (solid and dotted lines), including their creation (at the green stars), braiding (at the colored triangles), collision (in the yellow plane), and repulsion successively. **c** Tight-binding model. The three orbitals

of onsite energies $\omega_1(t)$ and $\omega_{2,3} = 0$ are coupled with hoppings $\pm \kappa$, $f(t)$, and $g(t)$. **d** Time dependence of $\omega_1(t)$, $f(t)$, and $g(t)$. **e** Time evolution of the nodal charges. **f** Momentum distributions of the band nodes and their charges at eight different moments, illustrating the braiding process in (**b**). The gray and cyan areas in $t = 4$ and 8 indicate the patch Euler classes $\chi = 0$ and $\chi = 1$, respectively. The colored arrows indicate the subsequent movements for the corresponding nodes.

adjacent gap, as described by the multiplication rule of QFCs. For example, after braiding with the gap-I node A($Q = -k$), the gap-II node B changes its charge (at the red triangle) from $-i$ to $+i$, i.e., $(-k) \cdot (-i) \cdot (-k)^{-1} = +i$[20,22]. Subsequently, the identically charged gap-II nodes B and B′ ($Q = +i$) cannot annihilate even after collision, and instead they bounce to another momentum line (Fig. 1b). Thus, the collision stability reflects the topology of a nodal pair in the same gap, and the topological phase transition can be induced by a nontrivial multi-gap nodal braiding.

We use a simple three-band square lattice model (Fig. 1c) to realize this non-abelian braiding process, which involves only one pair of band nodes in each gap[21]. The Hamiltonian of this model reads

$$H(\mathbf{k}, t) = \begin{pmatrix} \omega_1(t) & H_{12}(\mathbf{k}) & H_{12}^*(\mathbf{k}) \\ H_{12}^*(\mathbf{k}) & \omega_2 & H_{23}(\mathbf{k}, t) \\ H_{12}(\mathbf{k}) & H_{23}^*(\mathbf{k}, t) & \omega_3 \end{pmatrix}, \quad (1)$$

where the hoppings $H_{12}(\mathbf{k}) = -\kappa e^{-ik_x} + \kappa e^{-ik_y}$ and $H_{23}(\mathbf{k}, t) = 2f(t) + g(t)(e^{ik_x} + e^{ik_y})$. The onsite energies of the orbitals 2 and 3 are $\omega_{2,3} = 0$, the hopping strength between the orbitals 1 and 2 is $\kappa = 1$, whereas the onsite energy of the orbital 1, $\omega_1$, and the intra- and inter-cell hoppings between the orbitals 2 and 3, $f$ and $g$, vary with time, whose functional relations are given in Fig. 1d. The system features mirror symmetries $\mathcal{M}_{1\bar{1}}$ and $\mathcal{M}_{11}$, which protect the nodal point degeneracy in the (11) and (1$\bar{1}$) diagonals of the square BZ. The Hamiltonian can be made real-valued at all momenta after a unitary transform, thanks to the $C_{2z}T$ symmetry since $C_{2z} = \mathcal{M}_{1\bar{1}}\mathcal{M}_{11}$ (see Supplementary Note 1).

Based on the tight-binding model, we have calculated the full-BZ band structures at a series of representative moments (see Supplementary Fig. 1). An example can be seen in Fig. 1a ($t = 6$), which features gap-I nodes A (A′) and gap-II nodes B(B′). Figure 1e shows the time evolution of their QFCs, calculated with a consistent gauge selection (see Supplementary Fig. 2). Figure 1f sketches the nodal positions and their subsequent movements in the square BZ. At the initial time ($t = 0$), the three bands are energetically separated and host no band nodes. Next, a time-reversal-related nodal pair A(A′) of $Q = \pm k$ emerge at Γ-point ($t = 1$) and move in the (1$\bar{1}$) direction ($t = 2$). Similarly, at $t = 4$ and 6, another nodal pair B(B′) of $Q = \pm i$ nucleate at M-point and move in the (11) direction. The two pairs of nodes are braided reciprocally, resulting in sign-flipped QFCs for the nodes A′ and B($t = 7$). The identically charged nodal pairs will not annihilate after their collision, as exemplified by the gap-II nodes B and B′ (see $t = 7 \sim 10$). The collision stability of the nodal pair can also be characterized by a nontrivial Euler class defined over a patch around Γ-point ($t = 8$), in contrast to the trivial Euler class defined around M-point at its creation ($t = 4$). The (two-band) calculation of Euler class is straightforward and omitted here[21]. For brevity, hereafter we use the QFC to describe the whole three-band braiding physics.

## Ideal acoustic braiding metamaterials
We have designed ideal acoustic metamaterials to realize the introduced braiding effect, in which the time evolution is synthesized by using a sequence of acoustic samples. Figure 2a shows a photo of our acoustic metamaterial. As sketched in Fig. 2b, each unit cell contains three air cavities connected by narrow tubes. Physically, the cavity resonators emulate atomic orbitals, and the narrow tubes mimic hoppings between them[33–35]. We consider the lowest dipole resonance mode polarized along the length direction of each cavity. In particular, the structural parameters $L_\omega(t)$, $L_f(t)$, and $L_g(t)$ simulate the three variable parameters of the braiding Hamiltonian $H(\mathbf{k}, t)$, i.e., $\omega_1(t), f(t),$ and $g(t)$, respectively. Given that the resonance frequency is inversely proportional to the cavity length and that the coupling strength is (roughly) proportional to the cross sectional area of the narrow tube,

we present the time dependences of $L_\omega(t)$, $L_f^2(t)$, and $L_g^2(t)$ in Fig. 2c, which are piecewise functions like Fig. 1d (see details in Supplementary Fig. 4, Supplementary Table 1 and Supplementary Table 2). Figure 2d shows the simulated acoustic band structures at eight typical moments, which describes a nodal braiding equivalent to Fig. 1f. Excellent consistency is attained between the full-wave simulations and the model results (see Supplementary Fig. 5). Note that different from the bilinear dispersion around a general HSL node, the unique biquadratic (hybrid) line shape around a stable (unstable) HSP node can be directly used to visualize the collision stability of a nodal pair (see Supplementary Note 4).

The non-abelian nodal braiding unavoidably involves multi-band wavefunction entanglements. Below we elucidate that one can identify the stability of a colliding nodal pair through detecting the mirror eigenvalues of HSP Bloch wavefunctions. Specifically, along the HSL (1$\bar{1}$) or (11) two bands of opposite mirror eigenvalues can cross to form a pair of time-reversal-related nontrivial nodes[7,21], which may collide at the HSP Γ or M as the time evolution. Therefore, the collision stability of a nodal pair can be inspected from the mirror eigenvalues at given HSP: the nodal pair will not annihilate after collision if the eigenvalues of both mirrors are opposite for the two crossing bands; otherwise, the nodes will annihilate pairwise. To show this, we label the mirror eigenvalues of the HSP states on the band structures of $t = 0$, 6 and 10. As exemplified by the moment $t = 6$, the second and third bands carry opposite eigenvalues for both mirrors at Γ, which means that the colliding nodal pair in gap-II will not annihilate at Γ, but bounce from (11) to (1$\bar{1}$) direction (see next moments in Figs. 1f, 2d); on the contrary, the colliding nodes would pairwise annihilate at M since only the eigenvalues of $\mathcal{M}_{1\bar{1}}$ are opposite there, which coincides with their pairwise creation at $t = 4$ if we reverse the time evolution from $t = 6$ to $t = 2$. Similarly, one can predict the collision stability of the gap-I nodes: unstable at Γ (see $t = 0 \sim 2$) but stable at M (after $t = 10$). To visualize the HSPs' mirror eigenvalues, we present the eigenfield profiles in Fig. 2e for the moment $t = 6$. Intriguingly, the pressure patterns exhibit a nature of sublattice polarization, i.e., the orbital 1 and the other two orbitals cannot be occupied simultaneously, as enforced by the mirror symmetries of the system.

## Experimental characterization of acoustic nodal braiding
Our acoustic metamaterials were fabricated with using 3D printing technology. Each sample consists of 15 × 15 unitcells, and the total size is ~60.0cm × 60.0cm. To excite and scan the pressure field, the cavities are perforated with small holes (sealed when not in use) for inserting the sound source and detector. We place the sound source in the middle of the sample, and scan the real-space pressure distribution cavity by cavity. Finally, band structures in momentum space were achieved by performing 2D spatial Fourier transform (see Supplementary Fig. 6 and Supplementary Fig. 7).

Figure 3a shows the band structures (bright color) measured along the HSLs (11) and (1$\bar{1}$) of the square BZ. The experimental data, well-resolved in energy-momentum space, perfectly reproduce our numerical results (white dashed lines) except for the band broadening due to the dissipation and finite-size effect. In particular, our experimental data resolve clearly a biquadratic line shape around the stable gap-II node at Γ ($t = 8$), in contrast to the hybrid line shape sprouted from an unstable HSP node, i.e., the gap-I node created at Γ ($t = 1$) or the gap-II node created at M ($t = 4$), where the linear crossing happens along the HSL (11) or (1$\bar{1}$). For comparison, bilinear line shapes were checked for the nontrivial nodes at general HSL momenta (see Supplementary Fig. 8). Furthermore, in Fig. 3b we present the 2D spatial Fourier maps at the nodal frequencies for the instants $t = 4 \sim 10$, which directly visualize the creation, moving, collision and bounce of the gap-II nodes B(B′) in momentum space. This is another experimental evidence for the stable node ($t = 8$) evolved from the unstable one ($t = 4$) after braiding with the gap-I nodes A(A′). Thus far, we

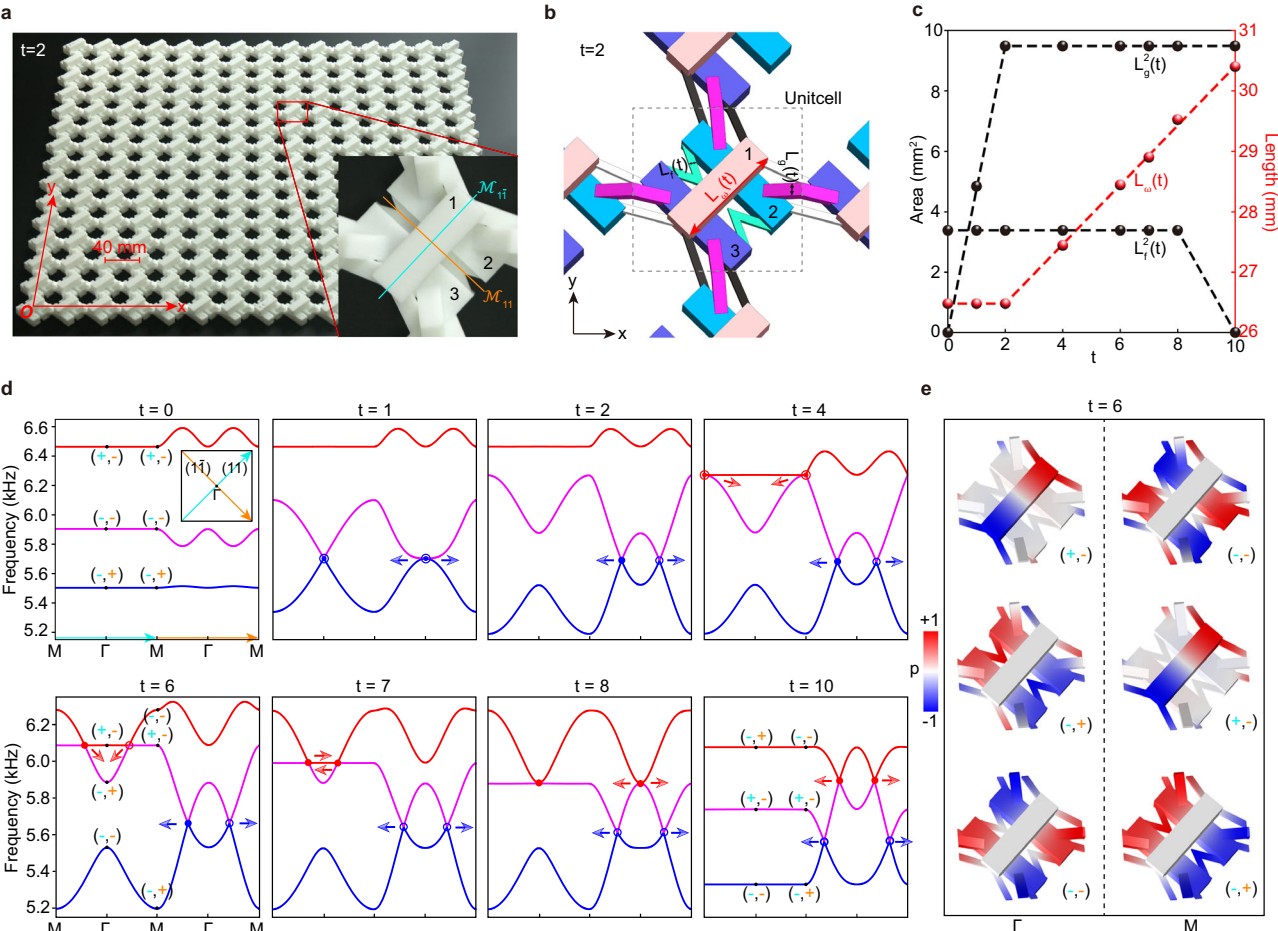

**Fig. 2 | Acoustic realization of elementary braiding model. a** A photo of our square-lattice acoustic metamaterial (with $t = 2$). The lattice constant $a = 40$ mm. Inset: an enlarged view of the unit cell with two mirrors labelled. **b** Unit-cell structure of our acoustic metamaterial, highlighted with three variable geometry parameters. **c** Time evolution of the geometric parameters. The discrete spheres correspond to our acoustic metamaterials at eight moments, and the dashed lines are fitted piecewise linear functions. **d** Band structures at different moments plotted along the diagonals of the square BZ (depicted in the inset of the $t = 0$ pannel). The blue and red arrows indicate the moving directions of the band nodes. In the cases of $t = 0$, 6 and 10, the eigenvalues of the mirrors $\mathscr{M}_{1\bar{1}}$ and $\mathscr{M}_{11}$ are labeled with $\pm$ at the HSPs, $\Gamma$ and M. **e** Eigenfield distributions of the HSP states at $t = 6$. The vanishing pressure fields ($p = 0$, white) in certain cavities exhibit a nature of sublattice polarization.

experimentally characterized a concise but nontrivial nodal braiding process, from a level of Fourier spectra.

The mirror eigenvalues of the HSP states, which can be used to conclude the stability of HSP nodes, were confirmed in our acoustic experiments. We focus on the case of $t = 6$ first. Specifically, we extracted the normalized wavefunctions at the HSPs from the measured 2D Fourier spectra, $|\phi_\mathbf{k}\rangle$ (with $\mathbf{k} = \Gamma, M$), and calculated their mirror expectation values, $\mathscr{M}_{1\bar{1}(11)} = \langle\phi_\mathbf{k}|\mathscr{M}_{1\bar{1}(11)}|\phi_\mathbf{k}\rangle$ (see Supplementary Note 5). Theoretically, $\mathscr{M}_{1\bar{1}(11)}$ approaching $+1 (-1)$ points to the mirror symmetric (antisymmetric) state of eigenvalue $+1 (-1)$. Figure 4a provides the measured expectation value spectra for the HSPs $\Gamma$ and M. As expected, for both cases, the spectra are peaked toward $+1$ or dipped toward $-1$ near each predicted eigenfrequency (dashed line), around which the sound intensity is maximized (see Supplementary Fig. 9). The mirror symmetries of the HSP states can also be directly inspected by the measured wavefunctions in Fig. 4b. Remarkably, the wavefunctions exhibit almost vanished pressure fields as predicted in Fig. 2e, which verifies the nature of sublattice polarization experimentally. Similar measurements were performed for the initial and final moments (see Supplementary Fig. 9). In particular, Fig. 4c, d present the mirror expectation values extracted for the three bands at $\Gamma$ and M. Consistent with the mirror eigenvalues depicted in Fig. 2d (see $t = 0$ and 10), our experimental data clearly unveil the band

inversions induced by multi-gap nodal braiding, which are inevitable intermediate steps to change the collision stability of nodal pairs. A more complete experimental characterization for the band inversion can be seen in Supplementary Fig. 10. The above experiments present evidence, from a deeper wavefunction level, for the band inversion and the consequent creation of stable HSP node, as the crucial signature of a nontrivial nodal braiding.

## Edge manifestation of non-abelian nodal braiding

Our $C_{2z}T$ symmetric system can also be used to study the braiding physics for each 2D sample by considering its subsystems formed by 1D loops of constant $k_{y(x)}$'s (given the periodicity of Brillouin zone), with $k_{x(y)}$ being a braiding parameter. Without losing generality, we focus on the 1D gapped subsystems of constant $k_x$'s but varying $k_y$, and the non-abelian band topology can be characterized by the quaternion charge $q$ defined for each loop. Interestingly, as exemplified with $t = 2$ (Fig. 5a), the loop charge $q$ can be related to the nodal charge by a simple quotient relation $Q = q_r/q_l$, where $q_r$ and $q_l$ describe the $k_y$-loops at the right and left sides of the node (see Supplementary Fig. 3). As such, the non-abelian topological transition occurs when the loop crosses a nontrivial node in the evolution of its $k_x$ index. This is illustrated in Fig. 5b by the $k_x$-dependent loop charge, which jumps at the

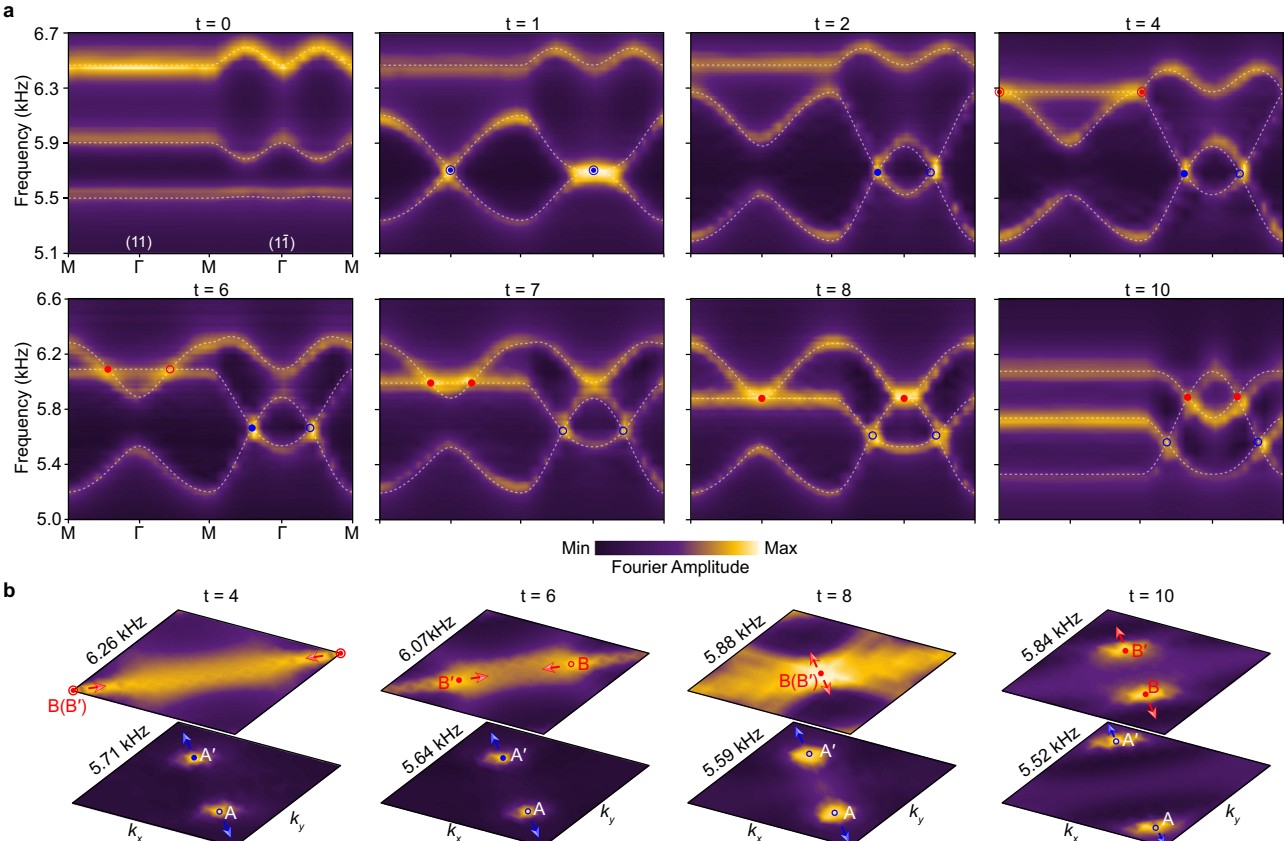

**Fig. 3 | Experimental observation of non-abelian braiding of acoustic band nodes. a** Measured bulk spectra along the HSLs for a sequence of acoustic meta-materials. All experimental data (color scale) excellently match the simulation results (white dashed lines). **b** Experimental characterization of the collision stability of the gap-II nodes B and B′ in the full square BZ.

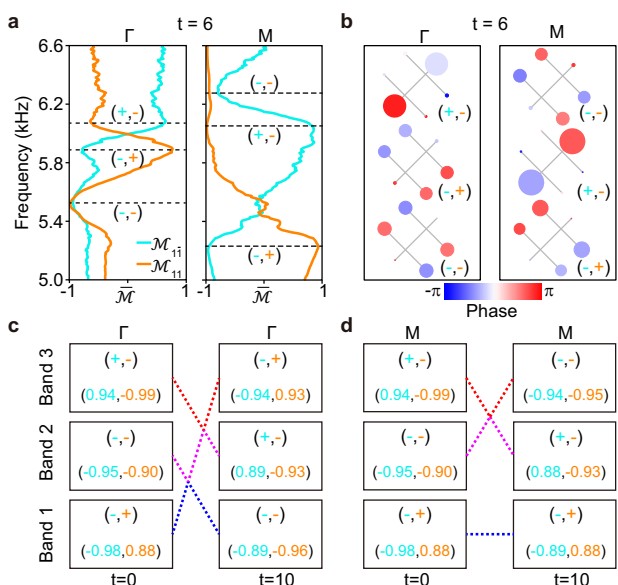

**Fig. 4 | Experimental characterization of mirror eigenvalues and band inversions. a** Expectation value spectra of the mirror operators $\mathscr{M}_{1\bar{1}}$ and $\mathscr{M}_{11}$ extracted for the HSP states at $t=6$. ± label the mirror eigenvalues of the HSP states at predicted eigen-frequencies (dashed lines). **b** Mapped field distributions for the HSP states, in which the size and color of the circles characterize the sound intensities and phases of the detecting points, respectively. **c**, **d** Band inversions manifested by the mirror expectation values (numbers) measured for the initial and final moments. Stable nodes can be concluded at Γ for the gap-II and at M for the gap-I, from the data of $t=10$.

nodal projections. More importantly, $q$ is closely related to the quantized Zak phases of the three gapped bands at the same constant $k_x$, and thus $q$ can be used to predict the presence of edge states in their two gaps[27], as summarized in Fig. 5c. (The case of $q=-1$ is an exception that goes beyond the Zak phase description[27], which does not happen in our system.) With these, we can explore the bulk-edge physics of each 2D sample from the perspective of braiding 1D subsystems, and track the bulk nodal braiding through the global evolution of the multi-gap edge states in different 2D samples. Similar edge manifestation of the bulk nodal braiding was reported in ref. [30].

The multi-gap bulk-edge physics was convincingly evidenced by our acoustic experiment (see details in Supplementary Note 5). Figure 5d–k show the measured edge spectra (color scale). All data capture well the simulations (color lines), benefiting again from the minimal nodal pairs in our braiding model. Notice that the edge responses are fully consistent with the theoretical predictions summarized in Fig. 5c, according to the $k_x$-dependent loop charges labeled in Fig. 5d–k. Specifically, for the moment of $t=0$, Fig. 5d exhibits no edge states because $q=+1$ everywhere; a similar result appears in Fig. 5e ($t=1$), except for the closure of gap-I at $k_x=0$ associate to the creation of a trivial node ($Q=+1$). For the moments of $t=2$ and 4, Fig. 5f and 5g show clearly topological edge states connecting the projected gap-I nodes, because the loop charge in this $k_x$ interval becomes $q=+k$ thanks to the split nontrivial nodes A and A′ ($Q=\pm k$). With the emergence of the gap-II nodes B and B′ ($t=6$), new edge states appear in this gap (Fig. 5h), consistent with the nontrivial loop charge $q=-i$ for the corresponding momentum range. Besides, there are no edge states appearing in the $k_x$ intervals between the gap-I and gap-II nodal projections, as a reflection of the loop charge $q=+1$. These facts

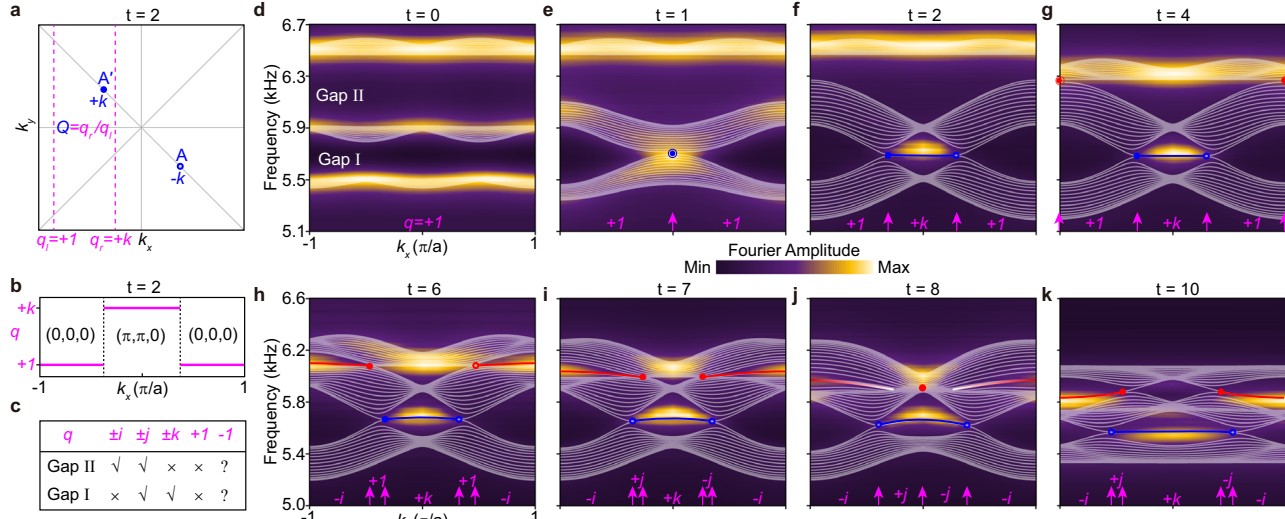

**Fig. 5 | Edge response of multi-gap non-abelian braiding. a** Quotient relation that relates a nodal charge ($Q$) and the non-Abelian charges defined for two constant-$k_x$ loops ($q_r$ and $q_l$). **b** $k_x$ evolution of the loop charge (dubbed $q$ in general) exemplified for the case of $t = 2$. Zak phases of the three bands (from the lowest to the highest) are indicated for different $k_x$ intervals. **c** A summary of the correlation between the loop charge and the presence (☑) or absence (☒) of the edge states in gap-I and gap-II. **d–k** Experimentally measured edge spectra (color scale) for a sequence of instants, compared with the associated simulation results (white and color lines for the bulk and edge, respectively). The arrows highlight the edge projections of the band nodes (circles), between which the corresponding loop charges are specified.

conclude that the gap-II nodes belong to the conjugacy class $\{\pm i\}$. For the following moments (Fig. 5i–k), new loop charges $q = \pm j$ emerge between the gap-I and gap-II nodal projections, so that both gaps host edge states in these momentum intervals. More importantly, from the time evolution of gap-II edge states, one may infer the emergence of a stable node ($Q = -1$) at $t = 8$: there are edge states emitted from the pairwise nontrivial nodes before and after their collision. In contrast, no edge state appears before the creation of an unstable node ($Q = +1$). All the above edge manifestations reflect faithfully the physics of the non-abelian braiding of bulk nodes.

## Discussion

Departing from a simple non-abelian braiding model, we have constructed ideal acoustic metamaterials that exhibits a concise but nontrivial nodal braiding process. Unambiguously, we have observed not only the evolution of bulk nodes with frequency-resolved spectroscopy in momentum space but also the topological transition and the nodal pair stability at the level of wavefunctions for the first time. The latter is particularly meaningful since essentially the non-abelian braiding physics aims to the wavefunction entanglements of multiple bands. Moreover, we have measured the edge responses to the bulk braiding and revealed the intricate correlation between the bulk nodal charges and the multi-gap edge states.

The fundamental rules governing the transition of non-abelian charges are crucial to understand the multi-gap topological physics. Our findings not only provide conclusive experimental evidence for the elusive non-abelian nodal braiding but also deepen our understanding of topological semimetals with noncommutative bands. By introducing additional non-Hermitian[36–38] or correlated physics[19,39], more charming topological braiding phenomena await to be discovered and manipulated.

## Methods

All full-wave simulations were performed by the finite-element based commercial software, COMSOL Multiphysics (pressure acoustic module). The photosensitive resin used for sample fabrication can be considered acoustically rigid, owing to the huge impedance mismatch with air (mass density $1.29 \text{kg/m}^3$ and sound speed $343 \text{m/s}$). To

simulate the bulk band structures of the acoustic metamaterials, Bloch boundary conditions were imposed in both $x$ and $y$ directions. The edge states were simulated with a strip of 15 periods in the $y$ direction, together with Bloch boundary condition applied to the $x$ direction.

In our bulk measurements, we placed a loudspeaker in the cavity of the middle unitcell of the whole sample, fixed a microphone (B&K Type 4138) as the phase reference, and used a needle-like microphone (B&K Type 4182) to detect the pressure signals of all cavities one by one. Similar measurements were performed to obtain the edge spectra, where the sound source was relocated to the middle unitcell of the bottom edge of the sample. Through spatial Fourier transform, we obtained the spectra in momentum space.

## Data availability

The data that support the plots in this paper and other findings of this study are available from the corresponding author upon reasonable request.

## Code availability

Numerical simulations in this work were all performed using the 3D acoustic module of a commercial finite-element simulation software (COMSOL MULTIPHYSICS). All related codes can be built using the instructions in the Method section.

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

## Acknowledgements

C.Q. is supported by the National Natural Science Foundation of China (Grant No. 11890701) and the Young Top-Notch Talent for Ten Thousand Talent Program (2019–2022); Q.Z. is supported by the National Natural Science Foundation of China (Grant No. 12104346). X.F. is supported by the National Natural Science Foundation of China (Grant No. 12004287). F.Z. is supported by the UT Dallas Research Enhancement Fund.

## Author contributions

C.Q. conceived the idea and supervised the project. H.Q. carried out the theoretical analysis and did the simulations. Q.Z. performed the experiments under the help of T.L. and X.F. H.Q., Q.Z., F.Z. and C.Q. analyzed the data and wrote the manuscript. All authors contributed to scientific discussions of the manuscript.

## Competing interests

The authors declare no competing interests.
