## [Peer Review File · Nature Communications]

Reviewer's comments:

Reviewer #1 (Remarks to the Author):

This manuscript presents a very interesting demonstration of the non-Abelian braiding of band nodes in multi-connected band systems using tunable acoustic metamaterials. With smart designs, the authors realize topological band nodes in two gaps with various frame charges. The tuning of the metamaterials yields the braiding of the band nodes while the frame charge changes and the underlying topology is manipulated. Moreover, the evolution of the topological edge states associated with such manipulation is revealed with consistent theory and experiments. The overall quality of this manuscript is very high. The topic is timely and certainly worth of study. I thus recommend the publication of the paper if the following minor issues are addressed:

1. The pictures in Figs 1-3 are very clear. However, figure 4 is not as informative as these figures. I suggest the authors to put more experimental data on the evolution of the mirror eigenvalues to support the evolution depicted in Figure 4 c and d.
2. The bulk-boundary correspondence in Ref. 27 are questioned in the community. However, the authors rather adopted the bulk-edge correspondence according to the Zak phase which is consistent with Ref. 30. Strictly speaking, the quaternion charges and Euler class are well-defined here in 2D systems (like in Ref. 30), while these quantities may not be well-defined in 1D (there is no smooth wave function manifold in 1D, but in 2D the loop definitions are well-defined). I thus encourage the authors to refer more to Ref. 30 which is a better context relating to their works.

Reviewer #2 (Remarks to the Author):

Three-band systems constitute the minimal setting for realizing topological phases characterized by non-abelian groups. In particular, quaternion charge characterization of band degeneracies in such multi-band systems underly their non-abelian braiding. While it is relatively easy to propose model Hamiltonians that implement such systems, finding their condensed matter realizations is difficult. Even more challenging is any realization of braiding protocols. Metamaterials (photonic, acoustic, etc.) provide a more straightforward means for solving the first task. In this particular case we are speaking of acoustic metamaterials manufactured by means of 3D-printing, which allow to reproduce the desired model Hamiltonians to a very high-degree of accuracy. However, the braiding protocol, which is the main claim of this paper, is realized in a simulated matter. Each of 8 points in time are described by a separate 3D-printed slab. In this case, it is perhaps an exaggeration to speak of the demonstration of braiding. The authors should tone down their claim and stress the fictitious nature of "time" in their

experiments. Other than that, I believe it is a high-quality work that deserves publication in Nature Communications.

Reviewer #3 (Remarks to the Author):

I. I find several statements in the introduction problematic and imprecise. Let me point some of these out.

1. The authors say ``Physical systems captured by this framework of “single-gap” topology are often characterized by abelian groups”.

This is imprecise since all topological classifications correspond to Abelian groups because the group composition rule here physically corresponds to stacking of quantum systems which is by default commutative.

2. The authors give a counter-example and cite references 19-22, in which allegedly non-Abelian phases have been described. This statement is incorrect since Refs 19-22 study phases that have non-abelian linking effects between line nodes in momentum space however the classification of such phases are still very much abelian.

3. In the introduction, the authors discuss classification of gapped topological phases while the paper pertains to gapless quantum systems, i.e, with for example line node degeneracies.

Due to these points I believe the introduction misrepresents the work and therefore should be re-written.

II. There is not much (if anything) new in the theoretical framework described in this paper as all of it has already been carefully described in Refs of Bzdusek et al. (Refs 20-22).

III. Acoustic metamaterials have been used in a variety of contexts to semi-classically emulate band physics and in particular topological band theory. This experimental platform therefore does not provide any new progress in the context of band topology. Given the fact that Refs 20-22 already describe in detail the phenomena presented in the paper and several works have already used a platform such as metamaterials to realize band insulators, topological phases, non-hermitian phases etc, it is not at all unexpected that non-Abelian braiding phenomenology can also be demonstrated in these platforms.

Due to the above reasons, I conclude that the paper under review does not represent sufficient scientific progress to warrant publication in the journal Nature communications.

Responses to the Referees

Reviewer #1 (Remarks to the Author):

This manuscript presents a very interesting demonstration of the non-Abelian braiding of band nodes in multi-connected band systems using tunable acoustic metamaterials. With smart designs, the authors realize topological band nodes in two gaps with various frame charges. The tuning of the metamaterials yields the braiding of the band nodes while the frame charge changes and the underlying topology is manipulated. Moreover, the evolution of the topological edge states associated with such manipulation is revealed with consistent theory and experiments. The overall quality of this manuscript is very high. The topic is timely and certainly worth of study. I thus recommend the publication of the paper if the following minor issues are addressed:

Reply: We appreciate the referee for his/her great effort in reviewing our manuscript. We also thank the referee for his/her high evaluation of our work.

1. The pictures in Figs 1-3 are very clear. However, figure 4 is not as informative as these figures. I suggest the authors to put more experimental data on the evolution of the mirror eigenvalues to support the evolution depicted in Figure 4 c and d.

Reply: We thank the referee for this valuable suggestion. In Fig. 4 we present the expectation value spectra of the mirror operators (Fig. 4a) and the associated field distributions (Fig. 4b) detected for the Γ and M states, together with the band inversion manifested by the mirror expectation values at the two high-symmetry points (Figs. 4c and 4d). Given the already *multi-layer* information in Fig. 4, we prefer to demonstrate the transition physics in the simplest way, i.e., providing only the mirror eigenvalues measured for the initial and final states. This concise way is more readable and retains the key information of the band inversion induced by the multi-gap nodal braiding. As a supplement, now we have provided more experimental data on the evolution of the mirror eigenvalues in Fig. S10 (see *Supplementary Information 5*).

2. The bulk-boundary correspondence in Ref. 27 are questioned in the community. However, the authors rather adopted the bulk-edge correspondence according to the Zak phase which is consistent with Ref. 30. Strictly speaking, the quaternion charges and Euler class are well-defined here in 2D systems (like in Ref. 30), while these quantities may not be well-defined in 1D (there is no smooth wave function manifold in 1D, but in 2D the loop definitions are well-defined). I thus encourage the authors to refer more to Ref. 30 which is a better context relating to their works.

Reply: We thank the referee for raising this point. As pointed out by the referee, the multigap

Zak phases play a key role in determining the (dis)appearance of edge states. Zak phase of each gap (i.e., bands below the gap) is well defined for a given edge momentum, which can change when the line of integration crosses a nodal point. From the bulk nodal-point configuration and its evolution, one can infer the evolution of edge states during the braiding process. Inversely, the evolution of the multigap edge states manifests the braiding of the bulk nodes. We fully agree with this logical chain first introduced in Ref. 30, which is pointed out more explicitly in our revised manuscript (page 10) as follows: “*The edge manifestation of the bulk braiding of the band nodes was first reported in ref. 30 based on the evolution of the momentum-resolved Zak phases.*”

In our work, we take this idea one step further. Our $C_{2z}T$ symmetric system can be used to study the braiding physics for each 2D sample by considering its subsystems formed by 1D loops of constant $k_{y(x)}$'s (given the periodicity of Brillouin zone), with $k_{x(y)}$ being a braiding parameter. Without losing generality, we focus on the 1D gapped subsystems of constant k_x 's but varying k_y , and the non-abelian band topology can be characterized by the quaternion charge q defined for each loop. Interestingly, q can be related to the nodal charge by a simple quotient relation [**Note 1**]. As such, the non-abelian topological transition occurs when the loop crosses a nontrivial node in the evolution of its k_x index. Moreover, q is closely related to the quantized Zak phases of the three gapped bands at the same constant k_x , and thus q can be used to predict the presence of edge states in their two gaps [**Note 2**]. With these, we explore the bulk–edge physics of each 2D sample from the perspective of braiding 1D subsystems, which reflects the bulk nodal braiding through the global evolution of the multi-gap edge states in different 2D samples. The prediction of the multi-gap bulk–edge physics is convincingly evidenced by our acoustic experiment. We note that the quotient relation between the quaternion charges of 2D nodes and 1D subsystems has not been reported elsewhere; it is a new result of our work, in addition to the realization of minimal non-abelian nodal braiding in metamaterials and its wavefunction-level evidence.

Note 1: *A careful study of the mentioned quotient relation is provided in Supplementary Information 2. Due to the gauge freedom of eigenstates, the sign of quaternion numbers could be flipped by choosing a different gauge. A recent theory [New J. Phys. 24, 053042 (2022)] has confirmed that the overall sign flipping does not cause any inconsistency in building the quaternion group. In our work we consider a synthetic 3D space formed by the 2D Brillouin zone and the parameter t , and we make the quaternion charges unique by fixing the gauge.*

Note 2: *The case of $q = -1$ is an exception beyond the Zak phase description, but it does not happen in our system. Conservatively, we choose not to use the term “bulk-boundary correspondence”.*

Reviewer #2 (Remarks to the Author):

Three-band systems constitute the minimal setting for realizing topological phases characterized by non-abelian groups. In particular, quaternion charge characterization of band degeneracies in such multi-band systems underly their non-abelian braiding. While it is relatively easy to propose model Hamiltonians that implement such systems, finding their condensed matter realizations is difficult. Even more challenging is any realization of braiding protocols. Metamaterials (photonic, acoustic, etc.) provide a more straightforward means for solving the first task. In this particular case we are speaking of acoustic metamaterials manufactured by means of 3D-printing, which allow to reproduce the desired model Hamiltonians to a very high-degree of accuracy.

Reply: We thank the referee for his/her great effort in reviewing our manuscript and for his/her pertinent assessment of our work. We agree with the referee that it is “*difficult*” to find condensed matter realizations for the model Hamiltonians and that “*even more challenging is any realization of braiding protocols*”. In fact, these motivated us to find a feasible platform toward the realizations. We elaborately designed acoustic metamaterials to “*reproduce the desired model Hamiltonians to a very high-degree of accuracy*”, as recognized by the referee.

However, the braiding protocol, which is the main claim of this paper, is realized in a simulated matter. Each of 8 points in time are described by a separate 3D-printed slab. In this case, it is perhaps an exaggeration to speak of the demonstration of braiding. The authors should tone down their claim and stress the fictitious nature of "time" in their experiments.

Reply: We thank the referee for this constructive suggestion. Following it, we have explicitly made the following revisions in our revised manuscript.

- In the abstract, “*By emulating the time with a sequence of acoustic samples, we experimentally observe...*”
- In the introduction (page 2) “*Here, we report ..., in which the braiding protocol is implemented by using a sequence of acoustic samples.*”
- In the main text (page 5) “*We have designed ideal acoustic metamaterials to realize the introduced braiding effect, in which the time evolution is synthesized by using a sequence of acoustic samples.*”

Other than that, I believe it is a high-quality work that deserves publication in Nature Communications.

Reply: We thank the referee for his/her high evaluation and recommendation of our work.

Reviewer #3 (Remarks to the Author):

I. I find several statements in the introduction problematic and imprecise. Let me point some of these out.

1. The authors say "Physical systems captured by this framework of "single-gap" topology are often characterized by abelian groups". This is imprecise since all topological classifications correspond to Abelian groups because the group composition rule here physically corresponds to stacking of quantum systems which is by default commutative.

Reply: We thank the referee for this comment. To address it, we have removed the word "often" in the revised manuscript.

2. The authors give a counter-example and cite references 19-22, in which allegedly non-Abelian phases have been described. This statement is incorrect since Refs 19-22 study phases that have non-abelian linking effects between line nodes in momentum space however the classification of such phases are still very much abelian.

Reply: We thank the referee for this comment. We agree that, while the linking between line nodes is non-abelian in Refs. 19-22, their *local* classification (i.e., for a single gap or a single line node) is abelian. However, once multiple band gaps are collectively considered, their *global* multi-gap band topology is characterized by non-abelian-group topological invariants. It is for this reason that the terms such as "non-abelian classification" and "non-abelian band topology" have been proposed in Refs. 19-22. Take Ref. 20 for example, "non-abelian band topology in noninteracting metals" is its title, and the abstract states "our analysis goes beyond the standard approach to band topology and implies the existence of one-dimensional topological phases not present in existing classifications." Similar statements have already been widely accepted and used in the community. Here are a few examples:

- Ref. 27: "Very recently, it is found that symmetry-protected topological phases can go beyond the Abelian classifications."
- Ref. 29: "Once multiple bandgaps are collectively considered, their coupling introduces richer physics that can make the classification non-Abelian."
- Ref. 30: "Rich non-Abelian topological phases are discovered during the evolution process."
- Ref. 31: "One of the most special configurations in non-Abelian system is the earring nodal link, composing of a nodal chain linking with an isolated nodal line, is signature of non-Abelian topology and cannot be elucidated using Abelian topological classifications."

To avoid any confusion, we have rewritten the first sentence of the second paragraph as follows: "Very recently, symmetry-protected multi-gap band topology have been predicted theoretically." After that, we introduce the non-abelian group characterization.

3. In the introduction, the authors discuss classification of gapped topological phases while the paper pertains to gapless quantum systems, i.e, with for example line node degeneracies.

Reply: In the introduction, we start with the familiar single-gap abelian band topology, then turn to the newly emergent multi-gap non-abelian systems, and finally narrow down to the gapless systems. Note that the topological invariant of a gapless system can be defined by its *fully gapped subsystem*; a simple example is Weyl semimetal that is characterized by the first Chern number. Here, the non-abelian quaternion charge is defined by a gapped k -space loop encircling the node. Therefore, we believe that our presentation is smooth and logical.

Due to these points I believe the introduction misrepresents the work and therefore should be re-written.

Reply: We thank the referee for his/her great effort in reviewing our manuscript. With the clarifications above and the corresponding revisions, we wish that the referee could find our current introduction reasonable now.

II. There is not much (if anything) new in the theoretical framework described in this paper as all of it has already been carefully described in Refs of Bzdusek et al. (Refs 20-22).

Reply: Regarding the significance and novelty of our work, we would like to point out the following three facts:

- Any compelling *experimental* evidence for non-abelian band topology is appealing and important, given that the study of this new class of band topology is still in its infancy. However, this is not easy to achieve, as pointed out by the second referee: “*While it is relatively easy to propose model Hamiltonians that implement such systems, finding their condensed matter realizations is difficult. Even more challenging is any realization of braiding protocols.*”
- For the first time, we constructed a sequence of *ideal* metamaterials with *minimal* band nodes to emulating the braiding effect. Unambiguously, we observed in our *experiments* not only the evolution of bulk nodes in momentum space but also the topological transition and the nodal pair stability at the *wavefunction* level.
- Also for the first time, we *theoretically* establish a quotient relation between the non-abelian charges of a 2D node and two 1D subsystems, *experimentally* unveil the bulk-edge physics of each 2D sample from the perspective of braiding 1D subsystems, and *experimentally* track the bulk nodal braiding through the global evolution of the multi-gap edge states in different 2D samples.

III. Acoustic metamaterials have been used in a variety of contexts to semi-classically emulate band physics and in particular topological band theory. This experimental platform therefore does not provide any new progress in the context of band topology. Given the fact that Refs 20-22 already describe in detail the phenomena presented in the paper and several works have already used a platform such as metamaterials to realize band insulators, topological phases, non-hermitian phases etc, it is not at all unexpected that non-Abelian braiding phenomenology can also be demonstrated in these platforms.

Reply: Here we respectfully disagree with the referee. It seems that the referee has some misunderstanding of the study of topological metamaterials in general. We would like to point out the following two facts.

- *Novelty in experiments.* While band insulators and several topological phases have been realized in metamaterial systems, unambiguous experimental evidence for the non-abelian nodal braiding remains elusive. Our experimental demonstration in a metamaterial system is *not a priori*. As an example, 3D strong and weak topological insulators (TI) were predicted in the same theory in 2017, however, while the first strong TI was experimentally observed (in $\text{Bi}_x\text{Sb}_{1-x}$) in 2008, the first weak TI was only experimentally identified (in $\beta\text{-Bi}_4\text{I}_4$) in 2019. Similarly, previous realizations of other topological phases in metamaterials neither imply that the non-abelian braiding can be easily realized in a similar system nor trivialize our experimental efforts. (This is clearly recognized by the second referee.) Indeed, our experiment was extreme challenging, because we needed to resolve the evolution of nodal points in energy-momentum space and demand the extraction of wavefunction information in Hilbert space. The latter is of prime importance as the braiding physics essentially aims to entangle multi-band wavefunctions. Now, both have been achieved in our elaborately designed acoustic metamaterials.
- *Broader impact.* In addition to examining topological band theories (that are elusive otherwise) and their universality across different systems, the study of topological metamaterials has been advancing the development of a variety of different types of metamaterials (such as photonic, microwave, phononic, acoustic, mechanical, and elastic systems) and enabling potential applications in each of their fields. Therefore, such study will have significant impacts on not only fundamental physics but also materials sciences and device applications.

Due to the above reasons, I conclude that the paper under review does not represent sufficient scientific progress to warrant publication in the journal Nature communications.

Reply: With our clarifications and revisions, we believe that the novelty, significance, and multidisciplinary impact of our work have met the high standards set by *Nat. Commun.* We wish the referee could agree with us now.

REVIEWERS' COMMENTS

Reviewer #1 (Remarks to the Author):

As far as I can see, the manuscript has been carefully revised and a detailed reply is also provided. My concerns are carefully addressed in the revised manuscript and in the reply, particularly regarding the non-Abelian topological charges and the bulk-boundary correspondence. Bulk-boundary correspondence in non-Abelian topological phases are still a topic with ongoing debates. Nevertheless, the authors give a part of the answer via a special approach. On the other hand, to characterize the bulk non-Abelian topology, it is key to detect the bulk Bloch wavefunctions which is realized only in recent years (see, e.g., arXiv:2205.03429

---the authors should also refer to this paper as well). One may note that such measurements have been seldomly been achieved in other physical systems. The direct measurements of bulk topological wavefunctions is hence of key importance in the study of topological physics and materials. In summary, I recommend the publication of this paper with minor revision.

Reviewer #2 (Remarks to the Author):

The authors have addressed my remark and made their manuscript more appropriate and clear to the reader. I recommend publication in Nature Communications.

Responses to the Reviewers

Reviewer #1 (Remarks to the Author):

As far as I can see, the manuscript has been carefully revised and a detailed reply is also provided. My concerns are carefully addressed in the revised manuscript and in the reply, particularly regarding the non-Abelian topological charges and the bulk-boundary correspondence. Bulk-boundary correspondence in non-Abelian topological phases are still a topic with ongoing debates. Nevertheless, the authors give a part of the answer via a special approach. On the other hand, to characterize the bulk non-Abelian topology, it is key to detect the bulk Bloch wavefunctions which is realized only in recent years (see, e.g., [arXiv:2205.03429](https://arxiv.org/abs/2205.03429)---the authors should also refer to this paper as well). One may note that such measurements have been seldomly been achieved in other physical systems. The direct measurements of bulk topological wavefunctions is hence of key importance in the study of topological physics and materials. In summary, I recommend the publication of this paper with minor revision.

Reply: We thank the referee for the recommendation to publish our paper in Nature Communications. Following the referee's suggestion, we have added the reference [arXiv:2205.03429](https://arxiv.org/abs/2205.03429) in our revised manuscript (see Ref. 32).

Reviewer #2 (Remarks to the Author):

The authors have addressed my remark and made their manuscript more appropriate and clear to the reader. I recommend publication in Nature Communications.

Reply: We thank the referee for the recommendation to publish our paper in Nature Communications.